# COVID-19 Sounds: A Large-Scale Audio Dataset for Digital Respiratory Screening

**Tong Xia**†*     **Dimitris Spathis**†*     **Chloë Brown**‡*     **Jagmohan Chauhan**‡◇

**Andreas Grammenos**‡*     **Jing Han**‡*     **Apinan Hasthanasombat**‡*     **Erika Bondareva***

**Ting Dang***     **Andres Floto***     **Pietro Cicuta***     **Cecilia Mascolo***

* **University of Cambridge,**◇ **University of Southampton**
† joint first authors, ‡ equal contribution, alphabetical order
`covid-19-sounds@cl.cam.ac.uk`

## Abstract

Audio signals are widely recognised as powerful indicators of overall health status, and there has been increasing interest in leveraging sound for affordable COVID-19 screening through machine learning. However, there has also been scepticism regarding the initial efforts, due to perhaps the lack of reproducibility, large datasets and transparency which unfortunately is often an issue with machine learning for health. To facilitate the advancement and openness of audio-based machine learning for respiratory health, we release a dataset consisting of 53,449 audio samples (over 552 hours in total) crowd-sourced from 36,116 participants through our COVID-19 Sounds app[1]. Given its scale, this dataset is comprehensive in terms of demographics and spectrum of health conditions. It also provides participants' self-reported COVID-19 testing status with 2,106 samples tested positive. To the best of our knowledge, COVID-19 Sounds is the largest multi-modal dataset of COVID-19 respiratory sounds: it consists of three modalities including breathing, cough, and voice recordings. Additionally, in this paper, we report on several benchmarks for two principal research tasks: respiratory symptoms prediction and COVID-19 prediction. For these tasks we demonstrate performance with a ROC-AUC of over 0.7, confirming both the promise of machine learning approaches based on these types of datasets as well as the usability of our data for such tasks. We describe a realistic experimental setting that hopes to pave the way to a fair performance evaluation of future models. In addition, we reflect on how the released dataset can help to scale some existing studies and enable new research directions, which inspire and benefit a wide range of future works.

## 1 Introduction

Since the outbreak of coronavirus (COVID-19) in early December 2019, there have been more than 211 million confirmed cases worldwide, including over 4 million deaths[2]. To combat the virus and control the transmission in time, various COVID-19 screening tools, such as PCR (polymerase chain reaction) [16], antigen [15], antibody tests [4], have been deployed. However, these approaches are

---

[1] `http://covid-19-sounds.org`
[2] `https://covid19.who.int/`

35th Conference on Neural Information Processing Systems (NeurIPS 2021) Track on Datasets and Benchmarks.

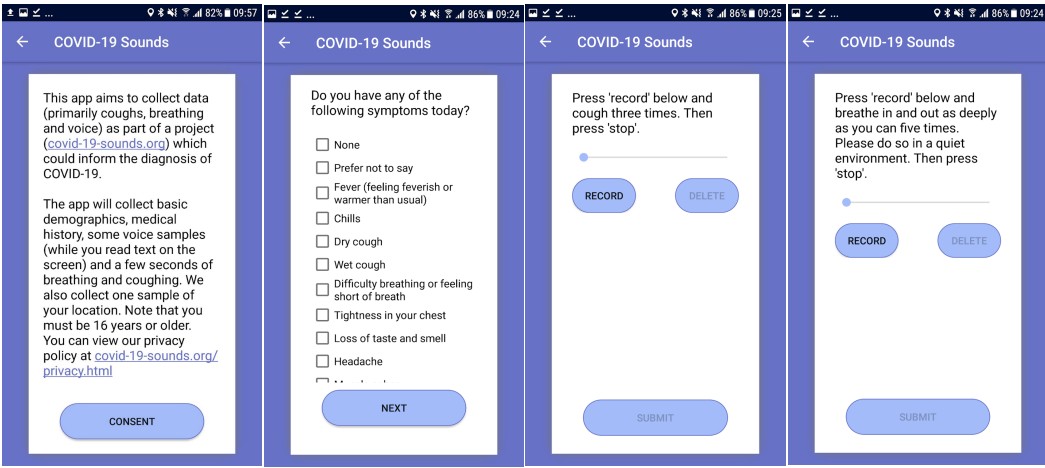

Figure 1: **Screens of the data collection app.** The users are asked to input their symptoms along with medical history, as well as to record breathing, cough, and voice sounds every couple of days.

invasive and relatively expensive, limiting the capacity for testing the public at scale. In this context, an affordable, non-invasive, fast, and ubiquitous COVID-19 testing tool would be a welcome addition worldwide.

Digital audio is an informative but underutilised modality in health [24]. Recently, researchers have started to explore whether respiratory sounds could be used for diagnosis of COVID-19 [13]. Stethoscope data from lung auscultation [21], coughs collected by phones [22], and speech recordings [18] have been analysed to distinguish COVID-19 patients from healthy participants. Quatieri *et al.* [32] showed that changes in vocal patterns could be a potential biomarker for COVID-19. However, comprehensive and transparent studies in this domain are rare, possibly due to the lack of large-scale, reliable, and open-source audio datasets. There has also been some level of scepticism expressed over these approaches [11, 19], possibly due to the lack of openness and reproducibility.

We hope the release of a large-scale audio dataset will help to pave the way to robust and open audio-based machine learning modelling for healthcare. This dataset is crowdsourced through the COVID-19 Sounds project, approved by the Ethics Committee of the Department of Computer Science and Technology at the University of Cambridge. During the data collection, participants are asked to report their demographics, medical history, and smoking status. In addition, they are required to report their COVID-19 test results, hospitalisation status, and symptoms (if any). To record respiratory sounds, participants are prompted to i) cough three times; ii) breathe deeply through their mouth three to five times; iii) read a short sentence on-screen and repeat it three times. Figure 1 illustrates some screen-shots of the app (Android version). After a year of data collection, 36,116 participants worldwide have contributed their data, and a dataset consisting of 53,449 audio samples (2,106 COVID-19 positive samples) is released.

To facilitate the utilisation of our dataset, we describe the data curation and statistics, and further present several benchmarks for two principal tasks: respiratory symptom prediction and COVID-19 prediction. The first task aims to explore the potential of sounds in detecting respiratory abnormalities manifesting as self-reported symptoms such as cough and fever, and the second task specifically investigates the capability of audio-based COVID-19 screening. For evaluation, two subsets with speaker-independent and demographic-balanced splits are used for the two tasks, based on which three benchmarks comparing traditional audio features to neural-based transfer learning are implemented, yielding a ROC-AUC up to 0.75 and 0.71, respectively. These results confirm the potential of audio-based machine learning models for these applications and the usability of our crowdsourced data. We hope the examined tasks can pave the way to simple but rigorous evaluation for future audio-based healthcare model developments. We additionally describe a plethora of applications like user authentication and smoking status detection, where our data can be utilised.

The contribution of this paper can be summarised as follows:

| Dataset | #Samples(#pos) | #Participants | Duration | Annotation | Sound Types | A | G | S | C | Other labels |
|---|---|---|---|---|---|---|---|---|---|---|
| *Virufy* [8] | 121(48) | 16 | 5 minutes | clinically validated | cough | ✓ | ✓ | ✓ | ✓ | smoking status, datetime |
| *Covid19-cough* [29] | 1,324(682) | n/a | 58 minutes | self-reported clinically validated* | cough | | | ✓ | | n/a |
| *Coswara* [36] | 2,030 (343)[†] | n/a | 50 hours | self-reported | breathing, cough, voice | ✓ | ✓ | ✓ | ✓ | smoking status, datetime, sites |
| *Tos COVID-19* [27] | 5,867(2,926) | 2,758 | 7.5 hours | clinically validated | cough | | ✓ | | | datetime |
| *COUGHVID* [26] | 27,550(1,156) | n/a | 35 hours | self-reported expert-labeled* | cough | ✓ | ✓ | ✓* | ✓* | datetime, countries |
| ***COVID-19 Sounds** (ours)* | 53,449 (2,106[‡]) | 36,116 | 552 hours | self-reported | breathing, cough, voice | ✓ | ✓ | ✓ | ✓ | smoking status, datetime, language |

Table 1: **Overview of audio datasets for COVID-19**. #pos: # of COVID-positive samples, A: age, G: gender, S: symptoms, C: comorbidities. n/a: not available, n/a for #Participants: no recurring participant ID to contribute multiple samples. *: partially labelled/validated. ✓*: only w or w/o. [†]: Cowara data collection is on-going and this statistic includes data before 16 August 2021. [‡]: 1,516 samples tested positive in the last 14 days, 534 in over 14 days, and 56 tested positive before without exact time, with a duration of 22.1 hours for all positive samples.

- We release a large-scale crowdsourced dataset of breathing, cough, and voice sounds with 53,449 samples from 36,116 participants[3]. This dataset covers various demographics and a wide spectrum of health conditions, and, particularly, including participants' self-reported COVID-19 testing status.

- We evaluate our dataset on two benchmark tasks and present effective baselines. Two subsets from the entire dataset are used, and, under the experimental setting for realistic performance evaluation, favourable results of ROC-AUCs of over 0.7 are achieved. This serves as a foundation for new state-of-the-art model developments for these two tasks.

- We discuss a range of promising applications where our dataset could be applied. Due to the scale of the audio and the extensiveness of the collected metadata, we believe this data has a great potential to expedite the research in the growing field of audio-based machine learning for health.

## 2 Related Work

In this section, we briefly put our work into context along with several other audio datasets for COVID-19 detection. In general, it is particularly time-consuming and challenging to collect comprehensive and high-quality data for health research. As COVID-19 is a newly discovered coronavirus, there are only a few publicly available audio datasets relevant to COVID-19. These existing datasets vary a lot in terms of types of respiratory sound recordings, the number of participants or samples, COVID-19 testing results, demographics, as well as other metadata. A comparison is summarised in Table 1.

**Data collection.** The variability across existing datasets is mainly attributed to the differences in data collection approaches; data collected from local hospitals usually have fewer samples (e. g., 16 participants in Virufy [8]), while online crowdsourcing tools are able to gather data at a large scale easily (e. g., over 27,000 recordings in COUGHVID [26]). Our app is one of the earliest launched crowdsourcing tools for COVID-19 sounds worldwide. Our collected data is large-scale, with three sound types and covering various metadata.

**Sound type.** Most existing datasets target cough sounds, since new and continuous cough is one of the main symptoms of COVID-19. Specifically, Virufy [8], COUGHVID [26], COVID19-cough [29], and Tos COVID-19 [27] only contain cough sounds. In addition to that, aiming at exploring the patterns typical to breathing and speaking sounds of COVID-19, Coswara dataset [36] also collected breathing sounds and voice sounds (sustained vowel phonation and counting digits from 1 to 20). Our collected data include breathing, cough, and voice sounds as well. But, different from Coswara, for our voice recordings, we chose induced speech, asking participants to read out a sentence off

---

[3]The data is released for academic research to academic institutions through a Data Transfer Agreement and not completely publicly due to the sensitivity of the data.

the screen multiple times, which has a longer duration and contains various phonemes. These three sound modalities enable both uni-modal and multi-modal model benchmarking.

**COVID-19 status**. In most datasets, especially crowdsourced, participants self-reported whether they have taken a test, and whether they have been diagnosed as COVID-19 positive. In contrast, in Virufy, Covid19-cough, and Tos COVID-19, clinically validated annotations via official PCR swab test are provided. Our labels come from self-reports. Although, admittedly, such methodology may result in reduced reliability of the labels, it is significantly more scalable for collecting large amounts of data outside the hospital environment.

**Metadata information.** In addition to the COVID-19 status, most existing datasets also provide other metadata, such as age, gender, the presence of relevant symptoms, the presence of comorbidities (e.g., asthma, COPD, pulmonary fibrosis), and smoking status. Likewise, we offer all the above labels. This information can be utilised to assess the effectiveness of a COVID-19 detection model on various population subgroups. Furthermore, it expands the usability of the dataset for other health-related research beyond COVID-19, such as respiratory condition prediction.

**Research impact.** Machine learning models have been developed using some of the aforementioned datasets [31], such as [3] using virufy and [33, 40] exploiting Coswara. Previously, two small curated subsets from our collected dataset were released as a part of our findings on COVID-19 classification from cough sounds [5] and from a combination of symptoms and voice [17]. The machine learning research community was fast to improve the model design based on the previously shared data [10, 43]. We also recently organised an open challenge in INTERSPEECH 2021, to detect COVID-19 from speech and cough [35]. This data release, however, is orders of magnitude larger, covering a global population, with more comprehensive audio annotations and metadata. This dataset will bring us closer to understanding whether an essentially free digital test for COVID-19 could become a reality. Also, most of the existing research trains and tests on a single dataset, while the availability of multiple datasets would enable more realistic models to be developed [1]. We hope our release can facilitate more research with comprehensive, transparent, and reproducible results, so as to benefit real clinical applications.

## 3 Dataset Description

In this section, we describe the features and unique characteristics of the dataset we release.

### 3.1 Reporting and Demographics

Every participant is assigned a unique anonymous participant ID upon installing the app[4] and is asked to fill in a sign-up survey with demographics and other metadata. After that, participants are asked every couple of days to record their sounds along with their current COVID-19 testing status. Given that we started collecting data in April 2020 when COVID testing was not universal, the large majority of the users has not been tested (34,302 samples). However, we know whether they were symptomatic or not. We distinguish between positive and negative tests, as well as whether this status was obtained in the last 14 days or earlier (1,572 recent + 924 previously positive samples). Especially for the negative status, we record if it is a recovery from a recent positive test or if the user had always tested negative (6,450 samples).

Regarding demographics, our data is slightly skewed towards male population (62%), the majority of the users are young to middle-aged adults (around 20-50 years old) and non-smokers, but there are smaller groups of ex- and current casual smokers. Our dataset is truly global, and our website (one platform) has recorded visits from 198 distinct countries. The statistics are displayed in Figure 2.

### 3.2 Audio Pre-processing

A sample (a data point) in our dataset consists of a breathing recording with three to five breaths, a cough recording with three voluntary cough sounds, and a voice recording based on a given sentence, repeated three times. An illustration of audio recordings is presented in Fig. 3.

---

[4]For web app users, a different ID is assigned with every visit.

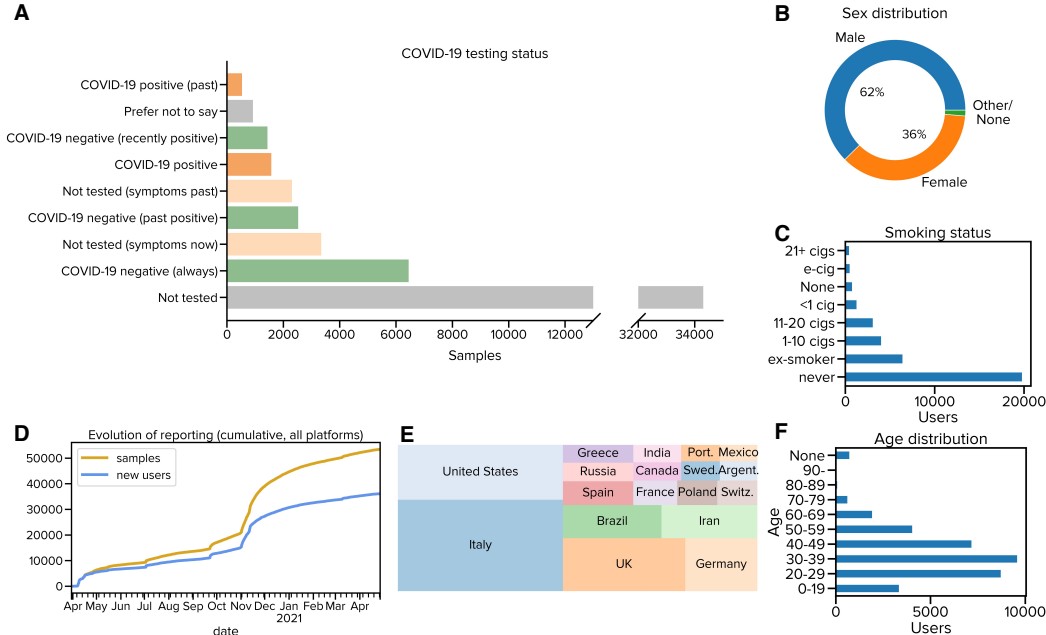

Figure 2: **Statistics of the COVID-19 Sounds dataset**. We group all samples into 9 main COVID-19 testing categories. Negative users are used as the control group in our COVID models **(A)**. Our sample skews slightly male (62%) **(B)**. The majority of users are non-smokers, but there are smaller groups of ex- and current casual smokers **(C)**. Our multi-platform app has seen stable growth in new and existing users over the past months, mostly helped by TV appearances and coverage in online media **(D)**. We plot a treemap of the top countries that contributed more than 190 unique users (all platforms). A big portion – mostly using the web app – chose not to disclose their location (11,042 not shown), while the rest of the users are based in Italy (4,698), the US (2,829), the UK (2,019), Germany (1,191), Brazil (1,016), Iran (987), Spain (522), Russia (407), Greece (393), and more **(E)**. The majority of users are between 20 and 50 years old, matching the population distribution of most western countries **(F)**.

As the dataset was crowd-sourced from various platforms, it contains a variety of audio files (i. e., .ogg, .m4a, .wav, and .webm) and sampling rates (i. e., 2.6% at 8KHz, 0.3% at 12KHz, 50.3% at 16KHz, 36.7% at 44.1KHz, and at 10.1% 48KHz). To simplify processing, we have converted all recordings to .wav files with original sampling rate preserved. Further, to ensure the reliability of our data, we developed an algorithm for automatic audio quality check by passing the audio through a pre-trained audio classification network YAMNet [20]. YAMNet is a deep convolutional network pre-trained on 521 audio event classes based on the AudioSet-YouTube corpus[5]. All recordings were re-sampled to 16KHz and were passed to YAMNet for sound classification to filter out noisy, silent, low-quality, and inconsistent recordings. To validate the effectiveness of YAMNet on our dataset, we checked a subset of recordings by manually-annotating them and compared the results to the prediction of YAMNet. With manually-assigned labels as the ground truth, YAMNet yielded an accuracy of 88.0% on 3067 audio recordings. Details can be found in Appendix (see Section A.1 in Supplementary Materials). As a result, starting from 53,449 samples (where each sample contains 3 recordings of different modalities), 43,518 breathing, 47,135 cough, and 49,382 voice recordings were identified as of sufficient quality. In total, 38,869 samples had sufficient quality across all three modalities.

### 3.3 Data Availability and Ethical Considerations

The data is sensitive, as voice can be deanonymised. Anonymised data will be made available for academic research upon requests directed to the corresponding email. Academic institutions will

---

[5]https://github.com/tensorflow/models/tree/master/research/audioset/yamnet

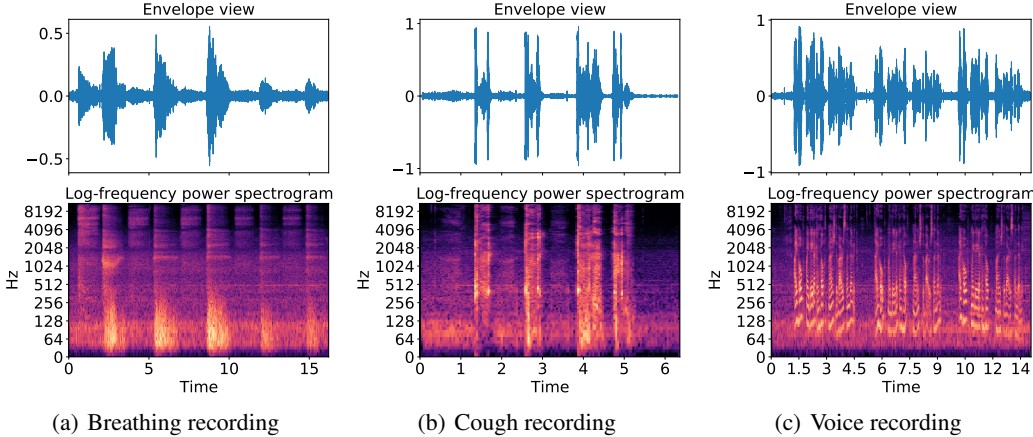

|  | | |
|---|---|---|
| (a) Breathing recording | (b) Cough recording | (c) Voice recording |

Figure 3: **Audio samples**. An example of recordings' waveforms and the associated spectrograms from a participant who tested COVID-19 positive within 14 days of the recording. The participant is a male aged over 30, with a smoking history, speaks English, had symptoms including wet cough, headache, and sore throat on the recording day.

need to sign a Data Transfer Agreement with the University of Cambridge to obtain the data. We have this agreement in place; please contact `covid-19-sounds@cl.cam.ac.uk` to obtain it. Once the document is signed, we will provide a download access to Google Drive where the data is stored.

The study was approved by the ethics committee of the Department of Computer Science at the University of Cambridge, with ID #722. For informed consent collection, our app displays a screen where we ask the for the user's agreement to participate in the study. Please note that the legal basis for processing any personal data collected for this work is to perform a task in the public interest, namely academic research. Under the Data Transfer Agreement, data recipients should keep the data confidential, and should not attempt to re-identify any individual from the data. They also need to inform and acknowledge us in the publications based on this dataset. Besides, they may use it for commercial purpose only if they get written consent from Cambridge in advance.

## 4    Benchmarks

### 4.1    Task and Model Development

In this section, we present benchmarks for two principal tasks that could be valuable to both machine learning and health research communities:

- **Task 1: Respiratory symptom prediction.** This task aims at exploring the potential of various sound types in predicting respiratory abnormalities, where the symptomatic group consists of participants who reported any respiratory symptoms, including dry cough, wet cough, fever, sore throat, shortness of breath, runny nose, headache, dizziness, and chest tightness, while asymptomatic controls are those who reported no symptoms.

- **Task 2: COVID-19 prediction.** This task aims to distinguish between participants who reported a COVID-19 positive status and those who reported testing negative. Note that the positive group may show no symptoms as there are many asymptomatic COVID-19 cases, while the negative group may show typical respiratory symptoms which are *not* caused by an active COVID-19 infection.

Three baselines are implemented for the above two tasks, with an SVM-based shallow model and two CNN-based deep learning models. For both two tasks, the input of the model is the audio sample consisting of breathing, cough, and voice recordings, and the output is a binary prediction. Thus, features extracted from breathing, cough, and voice recordings are fused in a concatenated manner, followed by shallow or dense classifiers. A brief introduction of the baselines is presented below, but more details can be found in Appendix (see Section A.2.2 in Supplementary Materials). We also

perform various ablations where a single modality from either breathing, cough, or voice sound is used in a uni-modal model.

- **OpenSMILE+SVM.** Following [17, 35], an established acoustic feature set is extracted with the open-source openSMILE toolkit [14] (384 features for each sound type) and is fed into an SVM classifier for classification.

- **Pre-trained VGGish.** We employ a pre-trained audio network (VGGish) to extract audio features automatically [20]. An average pooling layer handles the varying audio length, leading to a 128-dimensional feature vector for each sound type. Last, the resulting features (or embeddings) are used to train a classifier consisting of two fully connected and one softmax layers.

- **Fine-tuned VGGish.** Different from the aforementioned pre-trained VGGish baseline which fixes (or *freezes*) the parameters of VGGish, in this approach, we jointly fine-tune the backbone VGGish *and* update the fully connected layers, which ideally will result in more data-specific features for each task by "overriding" the pre-trained features.

## 4.2 Evaluation

The data for the experiments was selected according to the definitions of the above two tasks. To inspect the realistic performance, we conduct experiments following [11, 19]. Qualified samples are divided into speaker-independent sets for training, validation, and testing with a ratio of 7:1:2 (the validation set is used for hyper-parameter search and overall performance is assessed on the testing set). Demographics are balanced for each class and only English speakers are included to avoid potential language confounding[6]. Details can be found in Appendix (see Table 1 and 2 in Section A.2.1 in Supplementary Materials). Overall, 6,623 participants with 9,456 samples and 1,000 participants with 1,486 samples are utilised for the two tasks, respectively.

We report the following evaluation metrics which are widely used when assessing diagnostic tools: **ROC-AUC**, the area under receiver operating characteristic curve; **Sensitivity**, also called true positive rate or recall defined by $TP/(TP+FN)$ (in Task 1 the symptomatic group is regarded as the positive class); **Specificity**, also referred to as true negative rate formulated by $TN/(TN+FP)$. For all the metrics, we calculate 95% confidence intervals (95% CIs), using bootstrap re-sampling with 1000 bootstrap samples and replacement on the testing set [6]. As suggested by [28], the CIs can capture the variability of the accuracy when the model is deployed to different individuals. Overall, we hope the experimental setting serves as a foundation for new state-of-the-art model developments[7].

## 4.3 Results and Findings

Results are presented in Table 2 and 3. For Task 1, a ROC-AUC up to 0.75 (0.73-0.77) with a sensitivity of 0.70 (0.67-0.72) and specificity of 0.70 (0.67-0.72) is achieved. For Task 2, slightly lower performance of ROC-AUC up to 0.71 (0.65-0.76) with a sensitivity of 0.65 (0.57-0.72) and specificity of 0.69 (0.62-0.77) is obtained. We should note here that for Task 2 (COVID-19 prediction) some smaller datasets have reported superior predictive performance [37, 43], however, as we recently showed in [19], this is likely due to inherent biases and methodological decisions (such as unbalanced classes and reusing user groups for model tuning).

Deep learning outperforms traditional machine learning for both tasks, which is consistent with the literature showing deep features extracted from spectrograms of audio being more representative than handcrafted acoustic features [1]. Furthermore, fine-tuned deep models achieve better performance than the pre-trained but parameter-frozen models, suggesting the effectiveness of adjusting model parameters with task-specific data.

Cough is the best single modality, but the performance gain can be observed when we fuse breathing, cough, and voice, as they may provide complementary information for both respiratory abnormality and COVID-19 prediction. This finding suggests the superiority of multimodal audio models for

---

[6]We provide the splits in the released code for reproducibility reason, but we also recommend future works experimenting with different splits through balanced demographics to assess model robustness on different individuals.

[7]Code is publicly available on `https://github.com/cam-mobsys/covid19-sounds-neurips.git`

|  |  | ROC-AUC | Sensitivity | Specificity |
|---|---|---|---|---|
| **Breathing** | OpenSMILE+SVM | 0.60(0.58-0.63) | 0.47(0.44-0.50) | 0.67(0.64-0.70) |
|  | Pre-trained VGGish | 0.52(0.50-0.55) | 0.07(0.05-0.08) | 0.95(0.93-0.96) |
|  | Fine-tuned VGGish | 0.65(0.63-0.67) | 0.55(0.52-0.58) | 0.66(0.63-0.69) |
| **Cough** | OpenSMILE+SVM | 0.70(0.67-0.72) | 0.60(0.57-0.63) | 0.65(0.62-0.68) |
|  | Pre-trained VGGish | 0.66(0.63-0.68) | 0.67(0.64-0.70) | 0.53(0.50-0.56) |
|  | Fine-tuned VGGish | 0.74(0.72-0.76) | 0.70(0.67-0.73) | 0.68(0.65-0.71) |
| **Voice** | OpenSMILE+SVM | 0.63(0.60-0.65) | 0.56(0.53-0.59) | 0.62(0.59-0.65) |
|  | Pre-trained VGGish | 0.59(0.57-0.62) | 0.56(0.53-0.59) | 0.57(0.54-0.60) |
|  | Fine-tuned VGGish | 0.69(0.66-0.71) | 0.59(0.56-0.62) | 0.67(0.64-0.70) |
| **Fusion** | OpenSMILE+SVM | 0.74(0.72-0.76) | 0.65(0.62-0.68) | 0.69(0.66-0.72) |
|  | Pre-trained VGGish | 0.67(0.64-0.69) | 0.61(0.58-0.64) | 0.65(0.58-0.65) |
|  | Fine-tuned VGGish | 0.75(0.73-0.77) | 0.70(0.67-0.72) | 0.70(0.67-0.72) |

Table 2: Performance comparison for different modalities and models for Task 1. 95% CIs are reported in brackets. Best performance is highlighted in pink.

|  |  | ROC-AUC | Sensitivity | Specificity |
|---|---|---|---|---|
| **Breathing** | OpenSMILE+SVM | 0.56(0.50-0.61) | 0.38(0.30-0.45) | 0.70(0.64-0.77) |
|  | Pre-trained VGGish | 0.59(0.52-0.65) | 0.63(0.56-0.70) | 0.49(0.41-0.56) |
|  | Fine-tuned VGGish | 0.62(0.56-0.69) | 0.64(0.56-0.71) | 0.56(0.48-0.64) |
| **Cough** | OpenSMILE+SVM | 0.62(0.56-0.68) | 0.56(0.48-0.63) | 0.61(0.54-0.69) |
|  | Pre-trained VGGish | 0.62(0.56-0.68) | 0.69(0.61-0.76) | 0.45(0.38-0.53) |
|  | Fine-tuned VGGish | 0.66(0.59-0.71) | 0.59(0.51-0.65) | 0.66(0.59-0.73) |
| **Voice** | OpenSMILE+SVM | 0.52(0.45-0.58) | 0.43(0.35-0.50) | 0.62(0.54-0.69) |
|  | Pre-trained VGGish | 0.61(0.54-0.67) | 0.53(0.45-0.61) | 0.66(0.59-0.74) |
|  | Fine-tuned VGGish | 0.61(0.55-0.67) | 0.57(0.49-0.65) | 0.60(0.53-0.68) |
| **Fusion** | OpenSMILE+SVM | 0.64(0.58-0.70) | 0.54(0.47-0.62) | 0.67(0.60-0.75) |
|  | Pre-trained VGGish | 0.64(0.58-0.70) | 0.50(0.41-0.58) | 0.63(0.56-0.70) |
|  | Fine-tuned VGGish | 0.71(0.65-0.76) | 0.65(0.57-0.72) | 0.69(0.62-0.77) |

Table 3: Performance comparison for different modalities and models for Task 2. 95% CIs are reported in brackets. Best performance is highlighted in pink.

respiratory health, over existing approaches which use only coughs [26, 36]. Besides, a simple concatenation-based fusion was implemented, but different types of fusion approaches are worth exploring as future work.

We also conducted visualisation analysis on the fine-tuned VGGish model for Task 2. As illustrated in Figure 4, although there is no explicit boundary between the positive and the negative class, we can observe two clusters: red for positive and blue for negative (Figure 4(a)). For the incorrect predictions, we found that 9 of 18 asymptomatic positive samples were wrongly detected as negative (see Figure 4(b)), and 39 of 89 symptomatic negative samples were divided into positive group. Thus, we suppose that a participant's symptom onset and other comorbidities represent a real challenge to audio-based COVID-19 screening. The difficulty can also be validated by the predictive confidence distribution as shown in Figure 4(d): in the centre area where two classes mix, the confidence is lower than the two sides. This further suggests that incorporating uncertainty estimation in the automatic prediction will help to improve the reliability of the application [42].

Overall, under this realistic evaluation setting, sensitivity and specificity of 60-70% achieved by our benchmark models suggest promise in using sounds for affordable, non-invasive, and easy-to-access respiratory screening. Yet, we recognise that the current performance may not satisfy the safety requirements for clinical diagnostics.

## 5 Discussion

### 5.1 Data Limitations and Further Recommendations

Like in every crowdsourced dataset, the data has some caveats. Firstly, our data is crowdsourced and relies on the trustworthiness of the individuals' responses, especially the self-reported COVID-19 testing status. Also, it is worth noting that we did not collect the exact time/date of participants' COVID-19 testing, which means the sound recordings and the COVID-19 labels might have time shifts. For example, if a participant declared that they tested positive in the last 14 days, the real testing date remains unknown and could have been 12 days before the recordings are volunteered, by

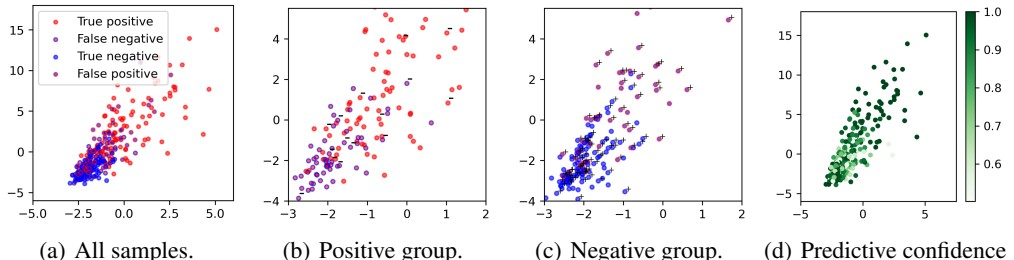

(a) All samples.     (b) Positive group.     (c) Negative group.     (d) Predictive confidence

Figure 4: **Visualisation analysis for Task 2.** We plot samples in the testing set by their VGGish embeddings through t-SNE. Each dot denotes one sample. In (a), (b) and (c), filling colour presents the label and edge colour presents the prediction, and in (d) colour shows the confidence of the prediction. (b) and (c) is a zoom-in of (a) with '-' denoting a asymptomatic positive sample, and '+' denoting a symptomatic negative sample.

which time the participant could have fully recovered. Yet, the approximate indication of a recent test seems to be sufficient to inform the machine learning models. Naturally, any conclusions from these models would welcome further clinical validation before large-scale usage.

As our apps have multiple versions in different languages and the data was collected globally, language and accent variance are also important to consider when using our data, particularly for voice. English is the majority language in our dataset (46.8%), followed by Italian (23.8%). If targeting a task with model generalisation to all languages, language bias might lead to unreliable performance. For instance, our previous study [19] shows that a cross-language deep learning model can achieve significantly different sensitivity/specificity on various language subgroups because of varied COVID-19 prevalence in language groups of the training data. Fairness for non-English speaking population is another concern, so we encourage more following studies to explore other languages and improve the model performance.

Our data collection framework includes multiple platforms and users' interpretations of instructions may vary, leading to some inconsistencies in the data like a different number of cough sounds in recordings. Besides, although we ask the users to record in a quiet environment, we could hear some background noise such as TV or radio in some recordings. Also, amplitude clipping on some cough and breathing recordings is evident because users speak too close to the microphone. Through the data quality check, we were able to exclude those non-usable samples mostly, but the impact of those inconsistencies in model performance remains unclear and needs further exploration.

All the aforementioned limitations should be taken into account when building diagnostic applications based on this dataset, in order to ensure that the model along with the results have a positive societal impact. The data collected is sensitive as it contains voices from participants which could be associated with the individuals if cross-examined with other datasets. This may potentially lead to linking of medical history or symptoms to specific individuals: our data is released with a data sharing agreement to protect against these operations, as described in detail in Section 3.3.

### 5.2 Potential Applications and New Research Directions

The tasks analysed in this paper show a glimpse of the potential of our collected audio data. There is a plethora of applications that can be empowered by such datasets. Some of these applications are discussed below:

**Symptom based COVID-19 prediction**: In addition to audio, other auxiliary information we collected could itself be of great value. Symptoms along with demographics can be employed for real-time COVID-19 tracking [38]. Comparing to previous studies [25], our dataset can not only be used for *an extra evaluation* of symptom-based COVID-19 models, but also *enables researchers* to investigate the interactions of medical history and smoking habits on manifestations of COVID-19.

**Biometric user authentication**: Audio modalities are increasingly used for user authentication. This does not only include voice, but also such modalities as coughing and breathing. For example, breathing acoustics [9] can be used as an authentication mechanism to gain access to mobile devices,

although with a very small number of users. Our dataset could help *scale* breathing-based user authentication and *enable a new verification* of cough-based user authentication as it includes different modalities from a large pool of users.

**Demographic prediction**: Accurate demographic predictions help modern voice assistants and chatbots to provide more targeted responses to users conversing with them. Audio from our dataset can be useful to do research related to predicting age group and sex from speech [39]. Moreover, our dataset provides *an additional opportunity* of exploring machine learning for cough/breathing based demographic perception.

**Respiratory disease detection**: Cough is considered a powerful biomarker for respiratory diseases. However, previous studies were based on very small-scale data. For example, [30] proposed a cough-based algorithm for an automatic diagnosis of pertussis using a dataset consisting of 21 samples. A number of works [7, 12] relied on the respiratory sound database released by ICBHI 2017 Challenge[8], but this dataset only consists of 920 annotated audio samples from 126 subjects. In contrast, our collected dataset contains cough/breathing sounds along with medical history at a larger scale. Specifically, 2,913 participants report to have been diagnosed with asthma, 297 are COPD patients, 322 have other lung illnesses, and 979 participants claim they suffer from other long-term chronic diseases. This could *facilitate* research in developing machine learning models based on different types of sounds.

**Smoking status detection**: Exposure to tobacco smoke affects throat tissues and causes inflammation to vocal folds, eventually leading to a variation in the speaker's speech sounds. Hence, many research works have explored the potential of using speech sounds to detect smoking status [2, 23]. Our collected dataset with 19,776 participants never smoking, 6,382 ex-smokers, and 8,718 smokers (1,245 participants smoke less than once, 3,995 participants smoke 1 to 10 cigars, 3,071 smoke 11 to 20 cigars, 407 smoke more than 20 daily) can further *accelerate* such research.

**Semi-supervised and self-supervised learning**: For audio-based COVID-19 detection, considering the large number of unlabelled samples (no COVID-19 testing status), making it more difficult to train deep neural networks, it is a natural next step to explore semi-supervised methods to boost the performance. Specifically, models could learn a robust representation on the non-tested participants and then apply this knowledge to the minority labelled set. A preliminary exploration has been done in [43], but more advanced approaches, such as incorporating symptoms into the semi-supervised learning framework, are worth exploring. Moreover, there is a number of generalised speech representation models [34, 41], but a pre-trained model tailored to respiratory health is missing. Self-supervised learning-based approach on our large-scale cough and breathing data may help to fill these gaps, resulting in general-purpose respiratory sound representations that can benefit a wide range of healthcare applications, including the tasks discussed above.

Other applications include cross-language model evaluation and country-level model comparison for the corresponding tasks. Overall, we think that our released audio dataset covers a wide spectrum of demographics and labels, which can benefit various machine learning studies and be used for a multitude of healthcare applications.

## 6 Conclusion

In this paper we describe the release of a large-scale audio dataset consisting of breathing, cough, and voice sounds crowdsourced from our COVID-19 Sounds app. This dataset covers a wide range of demographics and health conditions, and, most notably, it is the largest publicly available dataset for COVID-19 respiratory sounds. Two examined tasks with contributed benchmarks validate the usability of our data, as well as show the potential of exploring sounds for affordable health status prediction. We also discuss a plethora of applications for which our dataset could be used. The diversity of collected labels and the multiple data modalities could expedite research in the growing field of audio-based machine learning for health.

---

[8] https://bhichallenge.med.auth.gr/

## Acknowledgments and Disclosure of Funding

This work was supported by ERC Project 833296 (EAR) and the UK Cystic Fibrosis Trust. We thank everyone who volunteered their data.

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
