# OpenReview forum: "COVID-19 Sounds: A Large-Scale Audio Dataset for Digital Respiratory Screening"
_NeurIPS.cc/2021/Track/Datasets_and_Benchmarks/Round2 — NeurIPS 2021 Datasets and Benchmarks Track (Round 2)_

### Official Review · Reviewer_MAbU · 2021-09-02
**A relevant and well-motivated attempt in need of improvement**

**Rating:** 4
**Confidence:** 4

**Strengths:**

1. The paper is generally well-written and laid out in a logical manner, making it easy to digest and cross-reference sections of the paper. The illustrations are also of high quality.

2. If used correctly, the data has a potential for positive societal impact, for the fair assessment of feasibility of COVID-19 digital testing intervention methods, and the detection of ailments from the acoustic sound of cough, breathing and voice data.

3. The paper correctly identifies some potential sources of bias, and the authors have made a clear effort to report truthful figures, disregarding dubious claims made by previous efforts of the potential for COVID classifications from audio by other groups (e.g. 100% asymptomatic detection rate [Laguarta, Jordi et al.], 0.98 AUC-ROC classification rate by Andreu-Perez et al.).

4. The baseline models used with the data are simple, but reasonable, and sufficient for this purpose.




**Weaknesses:**

1. 34,302/53,449 samples of the dataset are without COVID 19 test labels: the paper title is clearly promoting this data as a means for COVID-19 detection, which makes this statistic surprising. There is thus a disconnect between the description of the data in the abstract, which does not mention or imply this. Furthermore, this makes Table 1 which compares several datasets, including their duration, misleading: supplying only the number of positive and negative PCR/other sources of labelled samples will be a fairer representation. Due to additional confounding factors such as language (and possibly the filtering of noisy samples with YamNet, which achieves approximately 82% accuracy on a key modality, the cough), the actual task of COVID detection is evaluated over only 1486 (2.8% of the total) samples. This further reinforces the disconnect between the motivation and claims of the paper and the way the dataset is used for baseline tasks.

2. The snippet of data provided to reviewers showcases a lack of documentation for metadata (as well as some metadata fields missing). The accessibility, hosting, and maintenance of the data is also a major concern. Additionally, there is no Datasheet provided (see further detail in the Documentation section).

3. At the time of review (02/09/21), there is no code supplied which is used to demonstrate performance over two principal research tasks. The visibility to the code in the paper is poor -- the only reference I could find to it is in line 616 of the supplementary materials with no indication of code availability anywhere in the main text. Code access is essential to the uptake of data in future, and it is difficult to provide a review without the opportunity to understand to what extent the code is documented, correct, and easy to use.

4. The additional uses cases mentioned in Section 5.2 can be interesting, but it is unlikely that people searching for COVID data will be interested in those tasks: to more effectively target a wider scope a change in title would be appropriate (or even change of key target applications and restructuring of the paper).


**Additional Feedback:**

I think this paper has great potential with major improvements to the work, and I would highly encourage the authors to continue working on this dataset.

Most importantly, clear documentation would significantly increase uptake and participation (and access to code). Several claims related to the potential/promise of the method based on the scores reported could use adjusting. Furthermore, a change in the overall title, scope and claims of the paper could increase the focus on the (majority) parts of the dataset which have not got a COVID label for alternate use cases as listed in the paper.

Further de-biasing and updating of metadata to include source audio format will be helpful to users. What is the bias in terms of audio formats used in training, validation and testing? Each audio compression format targets frequency bands in different ways, affecting spectral-based features. When using VGGish, the feature space will be filled with 0s for the lower sample rate signals. By resampling to wave but not stating the origin of the file, information about encoding is lost (e.g. which is `m4a`, which is `webm`, which is `ogg`?) It would be beneficial to include the format and sample rate in the metadata.

For all other suggestions for improvement please refer to the main body of text of the review.

# Post rebuttal
I'd like to thank the authors for their detailed responses. While the paper has undergone improvements as a result of the rebuttal, further concerns have arisen, and outstanding issues remain. My justification for these is given in https://openreview.net/forum?id=9KArJb4r5ZQ&noteId=lDyc_2QSZsV, and thus my score remains a 4.


**Clarity:**

The paper is overall very well written, and easy to navigate. The figures are of high quality and convey information well. There is however a lack of linking between the code, data, and metadata to the main text to enable researchers to more easily cross-reference the main text with the code and data.

**Correctness:**

To the best of my knowledge, the overall data collection, and engineering tasks are correct. It is however impossible to evaluate rigorously without access to the code and very limited access to data.

1. The statement
> Overall, these results reveal the great potential of employing sounds for respiratory condition prediction to achieve automatic screening through machine learning models

    is at odds with the figures, with reported results where some biases are still not investigated or understood, a test with 60-70 % both sensitivity AND specificity shows poor potential for real-world automatic screening applications.

2. For the baseline evaluations, though the method uses bootstrap re-sampling on the test data for CI intervals, different random seeds to partition data in 7:1:2 splits can be used to increase the reliability of the reported figures (while keeping demographic distribution similar as in Table 5). A similar bias table for input audio encoding/sample rate would also be useful, as it is unclear to what extent (if any), the difference in sampling rates will affect the outcomes.

3. A final point is that there are inconsistencies in the samples provided (different ways/rates of breathing due to their own interpretations of instructions)  and severe clipping on the cough and breath samples presented to the reviewers: has this created any problems in analysis or was this investigated to some extent?


**Documentation:**

1. The lack of documentation for this data is a major drawback of this submission. Reviewer samples that were made available are very poorly documented. There are fields with acronyms present which have not been defined, e.g. within `Smoking`: `ItOnce`, `Medhistory`: `copd`, `hbp`, and `Symptoms`: `pnts`. It is unclear what the total distribution of these is due to the data sharing agreement, but documentation for each field is essential. Furthermore, the documentation in the markdown readme and the paper supplement both contain the same typos within the voice check, breath check, and speech check sections. As an additional minor issue, the csv file is semi-colon delimited, which fails to display when opening by default in Excel 2016 (fine in Pandas if specifying , `delimiter=';'` with `read_csv()`).

2. Moreover, the supplementary materials do not include an appropriate Datasheet for the dataset as requested in the [Call for papers](https://neurips.cc/Conferences/2021/CallForDatasetsBenchmarks) e.g. [1](https://arxiv.org/abs/1803.09010), [2](https://arxiv.org/abs/1805.03677).

3. A further drawback is the lack of a convincing hosting and/or maintenance plan:
> Data will be shared through Google Drive once the DTA is signed.

    Is there a possibility to use a more secure/reliable platform for datasharing than Google Drive, perhaps adding a password protection or key encryption for access? This will be of particular concern over longer time periods (3 years+ for example).

**Ethics:**

The ethical implications are well covered, and I do not foresee any further issues to the best of my knowledge. Unfortunately the ethics also restrict the ease of access of this data, so I would wonder what differences to EPFL's COUGHVID dataset warrant restricted access and if it is worth removing a modality if it helps to more easily access this dataset in future?

**Relation To Prior Work:**

The overall structure of the related work section is very good, though I have concerns with the factual correctness and possibility of misleading statements being made.
1. The first point of contention is that the number of samples used to compare datasets (and thus the duration column) is unfair and misleading, as 34,302 samples of the dataset are unlabelled, and only 1,486 samples are used for COVID detection in the baseline tasks. In addition to containing crowdsourced labels only, with no known exact time of the test being taken, and furthermore automated tools required for quality screening, this significantly reduces the impact and usefulness of the presented dataset. For example, by number of positives, [27] exceeds this submission, and in addition is clinically validated.

2. Furthermore, when comparing metadata, the COUGHVID dataset is marked as not having comorbidities (C), however their data includes `respiratory condition`. Likewise, symptoms (S), which are marked as absent, are at least partially covered with `status` and `fever_muscle_pain`. For ease of access and useability, the EPFL COUGHVID dataset is far more convenient https://zenodo.org/record/4498364, with better formatted and explained metadata, though I cannot comment on the quality of the recordings without access to data.

**Summary And Contributions:**

This paper presents a crowdsourced dataset of cough, breath, and voice data for the primary purpose of COVID-19 classification from acoustic data. Data is accessed through a signed Data sharing agreement with the host university.

The data consists of 53,449 audio samples, of which 34,302 samples do not contain a label for COVID status. The two example tasks motivated for this paper are a) the detection of respiratory symptoms and b) COVID-19 prediction, using 9,471 and 1,486 samples for each task respectively. The paper describes three baselines for these tasks (handcrafted features + SVM, and two settings of the VGGish model). ROCs of a) 0.75 and b) 0.71 are achieved with the best-performing baseline model (pre-trained VGGish with trainable weights).

The contribution of this paper is to provide a large crowdsourced dataset to assist with COVID-19 classification, though other tasks are given. It also promotes a correct approach of using speaker-independent classification and attempts to mitigate bias in the data. It therefore inspires follow-up applications to train more useful models and/or validate existing data or models.

---

> ### Author Response · Authors · 2021-09-30
> **Rephrasing paper, improving documentation, and supplementing codes (II)**
>
> ### Relation To Prior Work:
> _1. The number of samples used to compare datasets is unfair and misleading_. Thanks for pointing out this. We revised Table 1, rephrased the claim of ‘largest dataset for COVID-19’ to ‘largest three-modality audio dataset for COVID-19’, and carefully adjusted the whole paper to make all claims rigorous and precise. For more details, please refer to our revised version. For benchmarking purposes, 1,486 samples were used in the COVID-19 detection task as an example to set up the experiments. More data samples can be included in the future as the data collection is still ongoing.
>
> _2. Comparing to COUGHVID dataset_. The reviewer is right that this dataset contains comorbidities (C) and symptoms (S). We have corrected this in Table 1.
>
> ### Documentation:
> _1. Document_. We are sorry for the unclear data instruction to the readers, so following your suggestion, we extended the document to include more details. Specifically, in the private URL where we shared examples, we changed the _meta_data.csv_ to a friendly format, so you view it on the Drive directly. Also, we submit a new data document named _Dataset document.xlsx_ elaborating on all fields in our data. Please refer to the excel sheet attached in the supplemental materials.
>
> _2. Datasheet_.  Thanks for this suggestion, following the template, we have supplemented a complete datasheet named _COVID19_Sounds_datasheet.pdf_. Please refer to the supplementary materials.
>
> _3. Data access and maintenance plan_.  We definitely understand the concerns of the reviewer, however as it is usually the case, this health dataset is sensitive: voice and cough could be deanonymized when integrated with other databases. Our University has therefore advised for a controlled sharing approach and we are not allowed to simplify or further streamline the data sharing process. Besides, we think we follow the call-for-paper guidance of this NeurIPS track as cited here:
>
> _“**Q: My dataset requires open credentialized access. Can I submit to this track?**_
>
> _A: This will be possible in the second round, on the condition that a credentialization is necessary for the public good (e.g. because of ethically sensitive medical data), and that an established credentialization procedure is in place that is 1) open to a large section of the public, **2) provides rapid response and access to the data**, and 3) is guaranteed to be maintained for many years”_
>
> Regarding maintenance, by submitting this paper, we aim to make our large-scale data public to facilitate more studies, and hence our team will respond promptly to all requests and make sure the data is available in the long term. We have already shared a small dataset for our previous study [5] with the same DTA more than 300 times, so we are experienced in data maintenance and sharing.
>
> _[5] C. Brown, J. Chauhan, A. Grammenos, J. Han, A. Hasthanasombat, D. Spathis, T. Xia, P. Cicuta, and C. Mascolo. Exploring automatic diagnosis of COVID-19 from crowdsourced respiratory sound data. In Proceedings of the 26th ACM SIGKDD International Conference on Knowledge Discovery & Data Mining (KDD), pages 3474–3484, 2020 (**citations: 114**)_
>
> ### Ethics:
> For the reason that 1) the risk of privacy leaking from the degradation is unclear as study [9] shows even breathing sound can be explored for user authentication, and 2) metadata is also sensitive, we are not allowed to simplify the data sharing process and to publicly release a degraded version. Data sharing should be restricted by the DTA, and once the DTA is signed, the data is easy to access, download, and use.
>
> _[9] J. Chauhan, Y. Hu, S. Seneviratne, A. Misra, A. Seneviratne, and Y. Lee. Breathprint: Breathingacoustics-based user authentication. In Proceedings of the 15th Annual International Conference on Mobile Systems, Applications, and Services, pages 278–291, 2017_
>
> ### Additional Feedback:
> _Audio format_. Thanks for pointing this out. We checked that the percentage of the different audio formats in training, validation, and testing is similar; In all sets, .m4a:.wav:.webm is close to 67%:32%:1%, so the original format would not introduce bias into the model. According to your suggestion, we have included the encoding format (this can be informed by the recording's name) and the original sampling rate. Please refer to the attached data document and the examples in the private URL.
>
> Thanks for your elaborate comments, which helped us improve the paper and data documents considerably.
>
> **A clearer document, a complete datasheet and code are supplemented. We rephrased large parts of the manuscript including the title to make the description of the data more precise and to improve its potential impact.**
>
> Thanks again for your feedback.

---

> > ### Comment · Reviewer_MAbU · 2021-10-03
> > **Reviewer conclusion**
> >
> > Thanks for formulating your reply -- I would advise doing this in future sooner than a few hours before the deadline. I personally didn't have access to edit reviews or comment from the reviewer console at this stage, and was only able to find a direct link via OpenReview's emails to allow me to do so. You may undertake changes during the rebuttal period after informing the reviewers what your plans are to address the criticisms.
> >
> > **W1+W4**: Thank you for acknowledging and correcting these inaccuracies - I am concerned about the factual accuracy of the remainder of the paper as I have not had the opportunity to check every claim in depth.
> >
> > **C1**: It seems the reference you give to show evidence exists for audio being useful for COVID-19 detection has a critical overall underlying message, and I would quote
> >
> > > These datasets will bring us closer to understanding whether the aspiration of an essentially free digital mass test for COVID-19 could become a reality.
> >
> > to imply it is more of an open question on its feasibility. This research question is happening in parallel to COVID-19 detection from CT scans, which one would expect to show promise given the relatively more advanced state of DL in vision vs audio, however, the field has been plagued with poor quality publications https://www.nature.com/articles/s42256-021-00307-0, and I believe this dataset you propose is not of sufficient quality for learning meaningful information regarding COVID-19 classification due to:
> >
> > * Absence of clinical verification
> > * Small number of positive samples + presence of too many confounding factors
> > * Requiring imperfect automated tools for the verification of recording quality
> >
> >
> > > Under the realistic evaluation setting, our model achieved sensitivity and specificity of 60-70%, indicating its potential for affordable, non-invasive, easy-to-access COVID-19 pre-screening outside the hospital, as this can help to better allocate clinical resources.
> >
> > A sensitivity AND specificity of 60-70% with an ROC of 0.71 - 0.78 would indicate not much better performance than a random classifier, and it is likely this figure is still too optimistic. I maintain therefore that *the statement is at odds with the figures*.
> >
> > **C2**: It is true that only *one* model will be deployed, but to test model robustness, one needs to evaluate how it would perform when trained and tested on different sources. A model to be deployed would be re-trained on all the available data anyway, so this comment does not address my main criticism here. The bootstrap re-sampling approach you quote states that
> >
> > > as the interval can capture the variability of the accuracy when the model is deployed to different individuals.
> >
> > which is exactly the point I was making for training and testing on different splits of the data (i.e. capturing the variability of the accuracy when deployed to different individuals). Instead, when looking through your code, I see that the train/validation/test data folds are pre-determined, so bootstrap re-sampling is performed on the test data only: **this does not capture the variability you could have captured by choosing different speaker-independent folds to use for your test data**. Furthermore, encouraging this evaluation method to for:
> >
> > > a foundation for new state-of-the-art model developments
> >
> > is dangerous as it would lead to models which are strong at performing on the (very limited due to the size, and lack of COVID verified samples) test set.
> >
> > ### Data hosting
> >
> > > ... and hence our team will respond promptly to all requests and make sure the data is available in the long term. We have already shared a small dataset for our previous study [5] with the same DTA more than 300 times, so we are experienced in data maintenance and sharing.
> >
> > This does not alleviate the concerns for hosting on Google Drive as opposed to a robust, version-controlled platform with credentialised access. Furthermore, what is the guarantee that this team will be available in more than five years' time when the ML COVID-19 research insurgency inevitably dies down?
> >
> >
> > ### Other comments
> >
> > Thank you for investigating sample rate, data clipping, and updating the documentation of the dataset.
> >
> > # Final verdict
> >
> > Upon engaging with the authors, going through the code and re-evaluating some statements, I acknowledge that the paper content is improved during the rebuttal, but the issues with the dataset are more concerning than initially conceived. I will therefore keep my score as 4 and update my review to acknowledge this.

---

> > > ### Comment · Reviewer_MAbU · 2021-10-03
> > > **Reviewer Conclusion (ii)**
> > >
> > > As the rebuttal period has closed I am unable to edit my original review, but I would like to mention to the area chairs that my score has not changed (see original review, author response, and reviewer response chain above).

---

> ### Author Response · Authors · 2021-09-30
> **Rephrasing paper, improving documentation, and supplementing codes**
>
> Thanks for your valuable comments. We respond to your concerns as below and have revised the paper accordingly.
>
> ### Weaknesses:
> _W1&4: Title, Table and Claims_. Thanks for pointing out these misleading aspects. We used ‘COVID-19 detection’ in the title because our initial motivation to develop such a data collection app is to battle COVID-19. We agree that at this moment, our samples with COVID-19 testing are not the majority, and instead, our dataset can be used for many other research purposes. Hence, according to your suggestion, we have revised our title to reflect that: _COVID-19 Sounds: A Large-Scale Audio Dataset for Digital Respiratory Screening_.  We also corrected the comparison in Table 1 and rephrased some claims in the paper to better describe the dataset. Please refer to the revised paper.
>
> _W1: Disconnect between the claim and the date used for baselines_. It is true that the data we used for Task 2 is a small subset of the entire COVID-19 data, mainly because we focus on providing a benchmark performance on _English_ speakers. For other users like Italian in our dataset, we encourage and welcome researchers to look into cross-language modelling.
>
> _W2: Document and datasheet_. We apologize for the unclear metadata documentation. We have now uploaded the new dataset document and datasheet (in supplemental materials) and strived for making the materials as complete and easy to follow as possible.
>
> _W3: Code availability_. Our code is now available in [Github](https://github.com/cam-mobsys/covid19-sounds-neurips.git). We made it private before and planned to open source it when the paper would have been accepted, but as requested at this review stage we hope it adds to our transparency and reproducibility efforts.
>
> ### Correctness:
> _C1: the statement is at odds with the figures_. Evidence suggests that COVID-19 produces identifiable features in infected individuals' speech, cough, and breath audio and progress is being made through the collection of several clinically validated datasets [11]. Hence, audio-based COVID-19 detection is promising. Under the realistic evaluation setting, our model achieved sensitivity and specificity of 60-70%, indicating its potential for affordable, non-invasive, easy-to-access COVID-19 pre-screening outside the hospital, as this can help to better allocate clinical resources. We do not claim that this technology is ready for diagnostic purposes, instead, we want to inspire future studies and enable research towards understanding the feasibility of such solutions in a transparent manner.
>
> _[11] H. Coppock, L. Jones, I. Kiskin, and B. Schuller. Covid-19 detection from audio: seven grains of salt.The Lancet Digital Health, 2021_
>
> _C2: baseline evaluations_. We echo the sentiment of the reviewer, however, we think that in real applications, only one model trained from the collected data will be deployed, thus it is not realistic to report the performance on various splits (cross-validation). Further, for the assessment of digital diagnostics, calculating confidence intervals with bootstrapping is more popular in the medical literature [28], as the interval can capture the variability of the accuracy when the model is deployed to different individuals.
>
> _[28] Platt, R. W., Hanley, J. A., & Yang, H. (2000). Bootstrap confidence intervals for the sensitivity of a quantitative diagnostic test. Statistics in Medicine, 19(3), 313-322_.
>
> _C2: the impact of the sampling rate_. In our proposed model, the pre-processing step involves resampling the recordings to 16KHz. In the benchmarks, only 2.6% and 1.8% of the used subsets had an initial sampling rate lower than 16KHz, and removing them from the experiments did not influence the overall performance. We added a note in the revised paper to clarify for future studies that if features are based on the high-frequency band, some examples can be discarded. Please refer to Appendix.
>
> _C3:  inconsistencies in the samples and clipping on the cough and breath samples_. We agree that variabilities of platforms, rates of breath/cough, users’ responses to the instructions, and audio quality all exist in our collected data. Still, our current model can inherently handle some of these inconsistencies: a sample, regardless of audio duration, is fed into the VGGish, and then average pooling is applied upon the time-axis, leading to a fixed-length feature vector that mainly captures the energy distribution in frequency, and last, this vector is used for classification. We have tried to use segmentation and only input segmented cough/breath/voice as input, but the performance was comparable. Hence, we present the model structure that involves no complex segmentation processing in this paper. In terms of clipping, the Yamnet which we employed for audio quality check can remove a large number of distorted samples. We plan to investigate its effect on our models in future work. We added a note to clarify that in the revised paper.

---

### Official Review · Reviewer_ZAVH · 2021-09-19
**A useful dataset**

**Rating:** 7
**Confidence:** 3
**Clarity:** The paper is well written.

**Strengths:**

1. The data scale is large. As claimed by the authors, it is the largest COVID-19 audio dataset collected so far.
2. The dataset is comprehensive in terms of demographics and spectrum of health conditions. Thus, it not only provides sufficient information for symptom and COVID-19 prediction, but also enables a wide range of new applications.
3. The dataset contains three modalities of audios, and ablation study has demonstrates their usability.


**Weaknesses:**

The data is released through a Data Transfer Agreement and not completely publicly due to the sensitivity of the data. The paper should describe more clearly how to apply for an access. Otherwise it would not be impactful for the research community.  I would like the raise my score if this concern can be well addressed in author response.

More baseline methods could be added to the experiments. In addition, detailed analyses and case study are appreciated for more insights.

**Additional Feedback:**

More baselines can be added to the benchmark, and further analyses can be made. For example, in what kind of scenarios the machine learning model will give the wrong prediction?

**Correctness:**

Both dataset construction and experiment are designed in a sound way.


**Documentation:**

There is sufficient detail on data collection and experiment reproducibility.

**Ethics:**

The dataset may course potential issues if it is be wrongly used or propagated. The own of this dataset should take care of this situation.

**Relation To Prior Work:**

In the related work section, the paper clearly discusses the difference of the presented dataset to previous works (Table 1). The dataset is larger and more informative than existing datasets.

**Summary And Contributions:**

This paper presents a large-scale audio dataset for respiratory symptom and COVID-19 detection collected from a crowd-sourcing app developed by the authors.  It also provides experimental results with three standard methods: OpenSMILE+SVM, Pre-trained VGGish, and Fine-tuned VGGish. As expected, VGGish performs the best. And further ablations prove the effectiveness of all three audio types, including breathing, cough, and voice. I believe this effort has remarked contributions to the research community, which enables fair comparison of existing models and facilitates development of new models. As the dataset contains comprehensive information about demographics and health conditions, it will also benefit many other applications, such as biometric user authentication, smoking status detection and respiratory representation learning.

---

> ### Author Response · Authors · 2021-09-30
> **The sensitivity of this dataset requires us to share it under data transfer agreement**
>
> Thanks for your valuable time. We would like to respond to your comments as below:
> ### Weakness:
> _Data access_. Thank you for pointing out the issue of unclear data access guidance. We definitely understand the concerns of the reviewer. However this health dataset is sensitive: voice and cough could be deanonymized when integrated with other databases, our University has therefore advised for a controlled sharing approach and we are not allowed to simplify or further streamline the data sharing process. The data should be only used for academic purposes and recipients will be restricted by the Data Transfer Agreement (DAT) to avoid illegal misuse.  This is also consistent with the call-for-paper guidance of the track as cited here:
>
> _“**Q: My dataset requires open credentialized access. Can I submit to this track?_**
>
> _A: This will be possible in the second round, on the condition that a credentialization is necessary for the public good (e.g. because of ethically sensitive medical data), and that an established credentialization procedure is in place that is 1) open to a large section of the public, 2) **provides rapid response and access to the data**, and 3) is guaranteed to be maintained for many years.”_
>
> We have a big research team maintaining the data and will make sure that we respond rapidly to any data access request, following the DTA. We have added a detailed description of the data sharing process in the revised paper. Our smaller datasets released previously with this DTA have already been shared with more than 300 institutions: this confirms that our framework has the potential to work well and reach many researchers.
>
> ### Further analysis:
> In this paper, we introduced the three most representative baselines, including handcrafted feature-based and deep learning-based methods, which we think should be sufficient as benchmarks. Also, since our main contribution of this paper is to provide a large-scale audio dataset to the research community, by defining tasks and proposing baselines, we aim to show the value of our data and how it can be applied to real-world problems. Following your suggestions, we added some visualisations/analyses to the revised paper (see Section 4.3), showing when our models fail. More baselines and case studies can be carried out in the future. Please refer to our revised paper.

---

> ### Comment · Reviewer_ZAVH · 2021-10-03
> **Data access concern has been solved.**
>
> I have revised my score since data access concerns have been solved after author feedback.

---

### Official Review · Reviewer_PbwN · 2021-09-20
**Audio dataset for the detection of COVID-19 symptoms**

**Rating:** 8
**Confidence:** 3

**Strengths:**

The main strength of the paper lies in the large scale of the dataset, which is significantly bigger than previous datasets proposed for the same tasks, and the presence of various labels (demographics, health conditions, etc.) that opens applications beyond the detection of COVID-19.

**Weaknesses:**

The main weakness of the contribution lies in the difficulty to assess properly the quality of the dataset. In particular:
1. The pre-processing step to remove audio samples with low quality is done automatically using an ad-hoc deep learning model, with an estimated accuracy of 88%, leading to a significant number of low-quality audio samples.
2. Labels are self-reported, and may not reflect the real health conditions of the participant.
3. The variability in to the recording conditions (different devices, background noises, etc.) is not discussed.

While some of these limitations are acknowledged by the authors, it is not clear how it could have a negative effect on the impact of machine learning models based on this dataset. In particular, albeit it is clearly mentioned that it is not suitable for clinical applications, the relevance of such dataset can be asked. A discussion of this point in the light of the recent debate about the relevance of machine learning for COVID-19 detection could be added to understand how this dataset may be relevant to solve actual problems.

The demographics of participants may also limit the universality of this dataset, with little participation from some countries, in particular with regards to their population. For example, is there any reason for Italy to be the most represented country (among those who responded)? It would be interesting to learn about the strategy employed by the authors to inform about the use of the application, and to reach underrepresented communities.

The imbalance in terms of demographics could also introduce biases that may limit the performance on certain group of the population and may result in serious fairness issues if the dataset serves as a basis for a clinical application. This aspect could be discussed more in details in the paper.

Finally, it is not clear what are the technical challenges associated with the dataset, and how they would require efforts ta deeper analysis of the data may help to provide the reader with factual elements about what could be the features of interest for the tasks. This could be done for example by computing basic speech characteristics (pitch, tone, etc.) in relation with symptoms, to guide the construction of relevant models. Another option would be to provide interpretable elements that guide the decision-making in the models built in section 4.

**Additional Feedback:**

Update after discussions with authors: I have updated my rating (from to 6 to 8) based on the answers provided by the authors on my review. My opinion is that the dataset is constructed in a sound way and has an adequate quality to be relevant for the machine learning community. The restricted access is justified by the nature of the data, and supplementary material is available to support the use of the dataset by other researchers.

**Clarity:**

The paper is well written, and the content is clear and well-organized.

The definition of tasks in Section 4 could be improved to avoid any ambiguity in the definition of problems, in particular regarding the input data and the outcomes expected.

**Correctness:**

The submission consists of a dataset and of two benchmarks. The dataset is constructed in a sound way, and has the potential to help the development of useful machine learning models. The benchmarks consist in two tasks related to the dataset. The level of details provided should be sufficient to reproduce the results, provided a working code is available.

The absence of available code alongside the paper is a limitation of the paper, as claims about the dataset and the results cannot be verified.

**Documentation:**

There is sufficient detail on the construction of the dataset. The dataset is not publicly available, and access to the dataset requires the establishment of a data transfer agreement. The procedure to set up this agreement is not currently described in in the paper nor in the project website, but only in the reviewers extra material. One main limitation seems to be in the requirement to guarantee of minimal level of security adapted for personal health data. It is not clear what it entails for the data recipient.

The process seems to be limited to academic institutions, and may seriously limit the use of the dataset by the machine learning community. A degraded version of the dataset (without voice, or with distortions to guarantee privacy) may be considered for public release.
The checklist provides minimal information on all necessary points, and may be further developed to expand the level of details of the dataset. It is unclear for instance whether additional data will be added in the futures and how these updates will be released.

**Ethics:**

Ethical considerations about the privacy of participants are discussed in the contribution, and resulted in the availability of the dataset only through a data transfer agreement, under conditions that are not precised.

An additional ethical concern that is not discussed is the possible repurposing of models built on this dataset for clinical applications, for instance for self-diagnosis. While it is mentioned that the dataset is not intended for clinical use, the nature of the tasks may lead to such applications, without the consent of the authors. The measures taken to limit this kind of misuse of the dataset may be detailed in the paper.

**Relation To Prior Work:**

The authors included a discussion of related works, and compared the characteristics of their dataset with previous ones.

**Summary And Contributions:**

This contribution proposes a dataset of audio sample of coughs, breathing, and voice recordings. The dataset includes more than 550 hours of audio from 36116, obtained through crowdsourcing. Each sample is accompanied by demographics about the speaker, and health conditions including COVID-19 symptoms. Connected to the dataset, two benchmarks are suggested: prediction of respiratory symptoms and of a COVID-19 infection. Baseline models with reasonable performances are proposed for those two tasks.

The contribution of the paper tackles a problem that could be of interest for the machine learning community, in particular concerning healthcare applications. Provided the dataset follows the methodology discussed in the paper, it is comprehensive enough to provide new challenges for machine learning experts, while leading to interesting results that could help in the development of automated techniques for the prediction of medical conditions.

---

> ### Author Response · Authors · 2021-09-30
> **Additional discussion supplemented in the revised paper (II)**
>
> ### Correctness:
>  _Code availability._ We provide the full code on Github https://github.com/cam-mobsys/covid19-sounds-neurips.git. We invite the reviewers to have a look and let us know of any questions that may arise. We have added a link also in the paper.
>
> ### Clarity:
> _The definition of tasks_. The input of the model is the audio sample consisting of breathing, cough, and sound recordings, and the output is binary. Specifically for task 1, we predict whether a user suffers from a  respiratory illness or not, and for task2, we predict whether a user suffers from COVID-19 or not. For these binary classification problems, we report both the sensitivity and specificity which are promising. We have improved the definition of the tasks in the revised version to make it clearer.
>
> ### Documentation:
> _No procedure to set up the agreement in the paper._  Thanks for pointing the data access issue out. We have explicitly introduced this in the revised paper.
>
> _What it entails for the data recipient._ As listed in the data transfer agreement, using the voice to re-identify participants and link meta to a specific person is not allowed. Data recipients can only use this data for academic research and should inform us when they are going to publish a paper. Hence, we can monitor what the data is used for.
>
> _Data access_. It is not impossible to share the data to be used for commercial purposes if recipients get the written consent of U of Cambridge in advance. Restricted by the Data Transfer Agreement (DTA) we have in place, the data should be only used for academic purposes. As the data was collected as part of an academic research project, the legal team of the University has deemed it necessary to restrict its use to such research.  This health dataset is sensitive because 1) the risk of privacy leaking from the degradation is unclear as study [9] show even breathing sound can be explored for user authentication, and 2) metadata is also sensitive, we are not allowed to simplify the data sharing process, and not allowed to publicly release a degraded version.
>
> _Other documents_. We have supplemented a complete datasheet and a detailed meta-data document. Code is also published. Please refer to the attachment. All the data has been prepared in Google Drive (access limited), and we will share the data with researchers who sign the DTA: in fact we have already started doing this for smaller subsets.
>
> ### Ethics:
> This dataset is collected for the research for COVID-19 and other health applications, with users’ consent as shown in Fig 1. The data transfer agreement restricts the recipient’s behaviour: data recipients should keep the data confidential, and should not attempt to re-identify any individual from the data. They also need to inform and acknowledge us in the publications based on this dataset. We have added this to the revised paper.

---

> ### Author Response · Authors · 2021-09-30
> **Additional discussion supplemented in the revised paper (I)**
>
> Thanks for your thorough review. Our responses are as below:
> ### Weaknesses:
> _W1: samples drop due to quality check_. We conducted an automatic quality check considering the large scale of our collected audio dataset. Yamnet achieved an accuracy of 88%, which we think can retain the majority of high-quality samples for further model development. Besides, Yamnet is being used by Google as one of the top-performing generic audio neural networks for more than 500 different classes. This accuracy is comparable to the cough detection accuracy proposed in COUGHVID [26]. Furthermore, if needed, for the 2106 COVID-positive samples, a manual check can be conducted to guarantee the quality and the number of usable samples.
>
> _W2: self-reported labels_.  This stems from the way we collected the data: crowdsourcing, which is efficient and allowed us to reach a considerable number of participants in a short period of time during the COVID-19 pandemic. In addition to COVID-19 labels, we also collect symptoms, which might be used to verify the COVID-19 labels. Developing machine learning models for health requires such large-scale datasets, and to compensate for the limitation of self-reported labels, our data can be jointly used with other small but clinically validated data for evaluation.
>
> _W3: The variability in the recording conditions_. Our data gathering framework includes multiple platforms: a web page, an Android app, and an iOS app, but the data collection process is the same, and our metadata contains the flatform type information. As for the background noise, we ask the users to record in a quiet environment, but for a small minority of the recording, we could actually hear some background noise such as TV or radio. Through the data quality check, we were able to successfully exclude those non-usable samples. We have added the related discussion to the paper in the revised version.
>
> ### Additional discussion:
>
> _The relevance between clinical applications and our dataset._ First of all, in our submission, we stated "Naturally, any conclusions from these models would welcome further clinical validation before large scale usage" and "Our dataset is released for academic use", but we didn’t mean "it is not suitable for clinical applications " and  "the dataset is not intended for clinical use". Instead, we actually suggest, for safety reasons, before deploying any model based on our data, a further clinical evaluation should be conducted. Second, as we discussed in the paper, "comprehensive and transparent studies in this domain are rare, possibly due to the lack of large-scale, reliable, and open-source audio datasets. There has also been some level of scepticism expressed over these approaches[11,19], possibly due to the lack of openness and reproducibility.", the relevance between clinical applications and our dataset is that we provide the opportunity to the machine learning community to develop and evaluate their models for healthcare applications.
>
> _The universality of this dataset._ We developed our app and localised it in English, Spanish, German, French, Dutch, Portuguese, French, Romanian, Italian, Hindi, Greek, and Chinese, which are available for free globally. We publicise the app worldwide and promote our study in different media (see our website‘s  press section). We can only offer a guess at why Italy is relatively more representative: 1) COVID-19 reached its first peak 2 months earlier in [Italy](https://covid19.who.int/region/euro/country/it) compared to the [UK](https://covid19.who.int/region/euro/country/gb). So after our preliminary study [5] being published in July 2020, Italian users responded promptly, contributing a considerable amount of samples.  2) There are a few sounds-based COVID-10 data collection apps, as summarised in Table 1, but to the best of our knowledge, our App is the only one in the Italian language. And 3) two investigators are from Italy and there was a strong media push in the country, which attracted many participants.
>
> _Demographic bias and fairness._ A great proportion (46.8%) of the participants are English-speakers. We think it is the language/accent rather than the location that may impact the audio-based COVID-19 screening. For non-English speakers, as you suggested, the performance needs to be further explored and the fairness issues should be carefully investigated. We have added these sections to the revised paper.
>
> _What are the technical challenges associated with the dataset_. One of the baselines OpenSmile+SVM conducted in this paper is an acoustic feature-based method, which was outperformed by the deep model. From the feature set, it is hard to find explainable, independent, and statistically distinguishable features for COVID-19. Hence, we suppose that acoustic changes caused by respiratory disease or COVID-19 are complex and sometimes subtle; capturing such changes adds to the rest of the technical challenges.

---

> ### Comment · Reviewer_PbwN · 2021-10-03
> **Response to the revision made by the authors**
>
> Thank you for your revised version, and the detailed answers you provided to my questions. and comments.
> I think the quality of the submission improved, and the availability of supplementary materials missing in the first version (data sheets, codes, etc.) makes it easier for researchers to use the dataset. The restricted access of the dataset is justified in my opinion, and I believe, thanks to the answers provided by the authors, that this will not significantly impact the relevance of the dataset for the research community. I will therefore update my review accordingly.

---

### Official Review · Reviewer_CvEz · 2021-09-21
**Expanding previous datasets**

**Rating:** 5
**Confidence:** 3
**Clarity:** The dataset and procedure are clearly…

**Strengths:**

1. Their dataset is the largest of its kind.
2. In addition to cough they have collected breathing, voice, and a few demographics and health conditions.


**Weaknesses:**

1. The number of COVID-positive samples is actually very small compared to the size of the dataset. Only 1,516 samples tested positive in the last 14 days. However, Tos COVID-19 dataset has 2,926 patients who are tested positive within 3 days.
2. In this dataset, the annotations are self-reported, which is scalable but not as reliable as clinically validated ones with CPR tests.
3. The data transfer process is way more complicated than for example Tos COVID-19, and some researchers might be denied access to the data.

**Additional Feedback:**

It would be nice to streamline the data transfer process.

**Correctness:**

The labels are all self-reported which raises some concern over the accuracy of the dataset.

**Documentation:**

A URL for reviewer access to the dataset is not provided, and their data transfer process is a logistical challenge that itself may take several weeks!

**Ethics:**

They have only mentioned that the study was approved by the ethics committee of Cambridge; however, it is not fully explained, and further implications are not discussed.

**Relation To Prior Work:**

Table 1 in the paper does a good job of comparing this dataset to the previously released dataset on the same topic.

**Summary And Contributions:**

The authors introduce a new dataset called COVID-19 Sounds which included more than 50000 audio samples from more than 36000 participants. Unlike the previously published dataset, it contains breathings and voices of the participants in addition to their coughs. Additionally, the authors evaluated simple baselines on their dataset on the detection of respiratory symptoms and COVID-19.

---

> ### Author Response · Authors · 2021-09-30
> **Response to data size, annotation, and access**
>
> Thanks for your review. We would like to respond to your concerns:
>
> ### Weaknesses:
> _W1: A smaller number of COVID-19 positive samples compared to TOS_.  Although we have fewer positive users, our data has several advantages compared to TOS: 1) We provide multiple modalities including cough, breathing, and voice rather than merely cough. This allows researchers to explore the impact of sound type on COVID-19 prediction, and as shown in Tables 2&3, multi-modal ML is beneficial over single inputs. 2) The reliability of the TOS dataset has not been validated by the research community: the original authors published an arXiv paper [27] with the evaluation section absent, and to the best of our knowledge, no study has yet been built upon that dataset. By contrast, the access, utilisation, and quality of part of our data have been recognised by many published studies, such as [5,17,34,42,43].  3) Although we acknowledge that most samples do not include COVID-19 testing results, our dataset includes a large number of samples with COVID-related symptoms, which could, in turn, enable the development of self-supervised or semi-supervised learning approaches.
>
> _W2: self-reported annotations_.  This stems from the way we collected the data: crowdsourcing, which is efficient and allowed us to reach a considerable number of participants in a short period of time during the COVID-19 pandemic. In addition to COVID-19 labels, we also collect symptoms, which might be used to verify the COVID-19 labels: using symptoms to diagnose COVID-19 and if the results are consistent with COVID-19 labels, the annotations are more reliable.  More importantly, developing machine learning models for health requires such large-scale datasets, and to compensate for the limitation of self-reported labels, our data can be jointly used with other small but clinically validated data for evaluation.
>
> _W3: data access_. We definitely understand the concerns of the reviewer, however as it is usually the case, this health dataset is sensitive: voice and cough could be deanonymized when integrated with other databases. Our University has therefore advised for a controlled sharing approach and we are not allowed to simplify or further streamline the data sharing process. The data should be only used for academic purposes and recipients will be restricted by the Data Transfer Agreement (DAT) to avoid illegal misuse.  This is also consistent with the call-for-paper guidance of this track as cited here:
>
> _“**Q: My dataset requires open credentialized access. Can I submit to this track?**_
>
> _A: This will be possible in the second round, on the condition that a credentialization is necessary for the public good (e.g. because of ethically sensitive medical data), and that an established credentialization procedure is in place that is 1) open to a large section of the public, 2) **provides rapid response and access to the data**, and 3) is guaranteed to be maintained for many years”_
>
> We have a big research team maintaining the data and will make sure that we respond rapidly to any data access request, following the DTA. We have added a detailed description of the data sharing process in the revised paper. Our smaller datasets released previously with this DTA have already been shared with more than 300 institutions: this confirms that our framework has the potential to work well and reach many researchers.
>
> ### Correctness:
>
> Please refer to our response to W2 self-reported annotations.
>
> ### Ethics:
>
> Thanks for pointing this issue out. We have extended the introduction of how the data collection and sharing are monitored by the ethics committee in the University of Cambridge. Please refer to Section 3.3 in the revised paper.
>
> ### Additional Feedback:
>
>
> Please refer to our response to W3 data access.
>
> Thanks again for your valuable time and constructive suggestions.

---

### Author Response · Authors · 2021-09-30
**To all reviewers: new documents, datasheet, codes, and the revised paper uploaded**

Dear reviewers,

We sincerely thank you for your valuable time and constructive comments, which helped us improve this paper considerably. We respond to your review point by point and hope to address all your concerns. In addition, we submit a revised paper with changes highlighted in blue. We also provide new supplements: the metadata document, the datasheet, a Github link with the requested code, and some snapshots to show how we organise the full dataset that will be shared upon acceptance. We hope our efforts make the paper and the data access clearer. Please check the new materials we submit and our responses below.

---

### Decision · Program_Chairs · 2021-10-09

**Decision:**

Accept

**Comment:**

The topic and dataset are of extremely relevance. However reviewers mention several drawbacks of the current version of the dataset and paper, including limited number of positive samples, lack of metadata and noise info, and the limitation of considering self-report annotations, among others. The authors did a great job discussing with the reviewers all critiques. Still some critiques hold but reviewers overall are more positive regarding the contribution. Overall the paper achieves the minimum required score to be accepted for publication at NeurIPS data track.